# Single Nucleotide Polymorphisms in the Vitamin D Metabolic Pathway as Survival Biomarkers in Colorectal Cancer

**DOI:** 10.3390/cancers15164077

**Published:** 2023-08-12

**Authors:** Carmen Pérez-Durán, Noelia Márquez-Pete, José María Gálvez-Navas, Yasmin Cura, Susana Rojo-Tolosa, Laura Elena Pineda-Lancheros, MCarmen Ramírez-Tortosa, Carlos García-Collado, María del Mar Maldonado-Montoro, Jesús María Villar-del-Moral, Cristina Pérez-Ramírez, Alberto Jiménez-Morales

**Affiliations:** 1General Surgery and Digestive System Service, University Hospital Virgen de las Nieves, Avda. de las Fuerzas Armadas 2, 18004 Granada, Spain; carmenperezduran@yahoo.es (C.P.-D.); jesusm.villar.sspa@juntadeandalucia.es (J.M.V.-d.-M.); 2Pharmacogenetics Unit, Pharmacy Service, University Hospital Virgen de las Nieves, Avda. de las Fuerzas Armadas 2, 18004 Granada, Spain; e.yacura@go.ugr.es (Y.C.); susanarojotolosa@gmail.com (S.R.-T.); lepinedal@unal.edu.co (L.E.P.-L.); carlosg.garcia.sspa@juntadeandalucia.es (C.G.-C.); mariadelmarmaldonadomontoro@gmail.com (M.d.M.M.-M.); alberto.jimenez.morales.sspa@juntadeandalucia.es (A.J.-M.); 3Biosanitary Research Institute ibs.GRANADA, Avda. de Madrid 15, 18012 Granada, Spain; cperezramirez87@ugr.es; 4Centro de Investigación Biomédica en Red en Epidemiología y Salud Pública (CIBERESP), 28029 Madrid, Spain; 5Cancer Registry of Granada, Andalusian School of Public Health, Cuesta del Observatorio 4, 18011 Granada, Spain; 6Department of Biochemistry and Molecular Biology II, Faculty of Pharmacy, Campus Universitario de Cartuja, University of Granada, 18011 Granada, Spain; mramirez@ugr.es; 7Pneumology Service, University Hospital Virgen de las Nieves, Avda. de las Fuerzas Armadas 2, 18004 Granada, Spain

**Keywords:** colorectal cancer, vitamin D metabolism, survival, single nucleotide polymorphisms, Caucasian, *VDR*, *CYP2R1*, *CYP27B1*, *CYP24A1*, *GC*

## Abstract

**Simple Summary:**

Colorectal cancer is one of the leading neoplasms in mortality worldwide. Recent studies suggest that vitamin D might influence the development of colorectal cancer and its prognosis. Furthermore, vitamin D activity might be influenced by the presence of single nucleotide polymorphisms in the genes involved in its metabolism. Thus, the aim of this study was to evaluate the influence of 13 SNPs in the vitamin D metabolic pathway on colorectal cancer survival. The results of the study showed that variants in *VDR*, *CYP24A1*, and *GC* genes are associated with a lower survival rate.

**Abstract:**

Several studies have suggested that single nucleotide polymorphisms (SNPs) related to vitamin D metabolism may affect CRC carcinogenesis and survival. The aim of this study was to evaluate the influence of 13 SNPs involved in the vitamin D metabolic pathway on CRC survival. We conducted an observational retrospective cohort study, which included 127 Caucasian CRC patient from the south of Spain. SNPs in *VDR*, *CYP27B1*, *CYP2R1*, *CYP24A1*, and *GC* genes were analyzed by real-time polymerase chain reaction. Progression-free survival (PFS) and overall survival (OS) were assessed. Cox regression analysis adjusted for metastasis, age of diagnosis, stage (IIIB, IV or IVB), ECOG score (2–4), lymph node involvement, adjuvant chemotherapy, and no family history of CRC showed that the *VDR* ApaI (*p* = 0.036), *CYP24A1* rs6068816 (*p* < 0.001), and *GC* rs7041 (*p* = 0.006) were associated with OS in patients diagnosed with CRC, and *CYP24A1* rs6068816 (*p* < 0.001) was associated with PFS adjusted for metastasis, age of diagnosis, stage (IIIB, IV or IVB), ECOG score (2–4), lymph node involvement, adjuvant chemotherapy, and no primary tumor resection. The rest of the SNPs showed no association with CRC survival. Thus, the SNPs mentioned above may have a key role as prognostic biomarkers of CRC.

## 1. Introduction

Colorectal cancer (CRC) is one of the leading causes of cancer mortality worldwide (9.4%) [1]. It has a prevalence of 11.9% and an incidence of around 8%, and it is the most common cancer after lung, breast, and prostate cancer [1,2]. According to the latest cancer statistics, there are expected to be more than 153,020 new cases and 52,550 deaths from CRC in the United States in 2023 [2]. 

CRC involves the uncontrolled growth of malignant cells in the colon and rectum, with the ability to invade other tissues and create metastases [3]. The main risk factor for CRC is age, with 90% of cases being diagnosed in people over the age of 50 [3]. Similarly, overweight and obesity, smoking, excessive alcohol consumption, and physical inactivity, as well as consumption of certain foods such as processed meats, could influence the development of CRC [4]. Scientific advances are focusing on looking for new predictive biomarkers related to CRC risk and survival [5,6]. With respect to this, recent studies have shown a strong association between blood vitamin D levels and the risk of developing CRC [7,8].

Vitamin D is synthesized endogenously from 7-dehydrocholesterol after exposure to sunlight or via direct ingestion in the diet [9]. It is subsequently hydroxylated in position 25 by action of the hepatic enzyme 25-hydroxylase (CYP2R1), converting it to calcidiol (25(OH)D), the main circulating form in the blood. Next, the enzyme 1α-hydroxylase (CYP27B1) performs a second hydroxylation (in position 1) and converts the 25(OH)D to 1,25-dihydroxyvitamin D [1,25(OH)_2_D_3_], either in the kidney, where it is released into circulation, or in specific target organs [9]. Circulating 1,25(OH)_2_D_3_ is broken down by CYP24A1 into calcitroic acid and other water-soluble products that are inactive and are excreted in the bile or urine [9]. The two active metabolites of vitamin D bind to the vitamin D binding protein (VDBP or DBP), previously known as group-specific component (GC), facilitating its transport [10]. In the target tissues, 1,25(OH)_2_D_3_ interacts with the vitamin D receptor (VDR), which binds to the retinoid X receptor (RXR), forming a heterodimeric complex; this is translocated to the nucleus, where it interacts with the VDR response elements (VDRE), at multiple genomic loci [9]. For all these reasons, VDR is key to the performance of the biological actions of 1,25(OH)_2_D_3_ [8]. Furthermore, vitamin D has proven to have an inhibitory effect on angiogenesis and to arrest the G_0_/G_1_ cell cycle, induce apoptosis, increase cell differentiation, and inhibit various signaling pathways in tumor cells [11]. It is estimated that vitamin D contributes to the expression of 3–5% of genes, notably including those related to cancer (for example, *MYC*, *BAK1*, *BAX*, *cyclin D1/2/3*, *GFM1/2*) (Appendix A) [6,11]. 

The involvement of this biomolecule in the expression of tumor genes could be linked to the action of the VDR, since high expression of the *VDR* gene has been observed in the small intestine and colon [7]. Thus, vitamin D would be involved in the initiation and development of carcinogenesis in both organs [5,6,9]. Therefore, single-nucleotide polymorphisms (SNPs) have been described in genes involved in vitamin D metabolism (*VDR*, *CYP2R1*, *CYP27B1*, *CYP24A1*, and *GC*) which could substantially affect serum vitamin D levels or modify its action [12,13]. Moreover, there is scientific evidence relating the presence of SNPs in the genes that participate in vitamin D metabolism to CRC survival [14,15,16].

On the basis of the foregoing, we have carried out this study to assess the SNP-type variants in the genes involved in the vitamin D metabolic pathway, with an allelic frequency higher than 1%: *GC* rs7041 (A > C), *CYP2R1* rs10741657 (A > G), *CYP27B1* rs10877012 (G > T), rs4646536 (A > G), rs3782130 (G > C), and rs703842 (A > G), *VDR* rs1544410 (BsmI) (C > T), rs731236 (TaqI) (A > G), rs7975232 (ApaI) (C > A), rs2228570 (FokI) (A > G), and rs11568820 (Cdx2) (C > T), and *CYP24A1* rs6068816 (C > T) and rs4809957 (A > G), as recorded in the dbSNP database, with the object of ascertaining their impact on the survival of Caucasian patients from southern Spain diagnosed with CRC [17].

## 2. Materials and Methods

### 2.1. Study Design 

We conducted an observational retrospective cohort study.

### 2.2. Ethic Statement

The study was carried out with the approval of the Ethics and Research Committee of the Sistema Andaluz de Salud (SAS: Andalusian Health System) in accordance with the Declaration of Helsinki (code: 1322-N-20). The subjects participating in the study signed an informed consent for the collection and genetic analysis of the saliva samples and their donation to the Andalusian Public Health Service’s Biobank. The samples were identified by alphanumeric codes.

### 2.3. Study Population

The study was conducted at the Hospital Universitario Virgen de las Nieves (HUVN) in Granada, Spain. A total of 127 Caucasian patients originally from southern Spain were diagnosed with colon cancer in the Oncology Department of the HUVN between 2009 and 2022 and in follow-up until February 2022. The inclusion criteria for the study subjects were age over 18 years, diagnosed with colon cancer (stages I–IV), with adequate organ function, with disease assessable by computed tomography, with no previous treatment and available clinical data. The patients were treated in accordance with the National Comprehensive Cancer Network (NCCN) guidelines [18].

### 2.4. Sociodemographic and Clinical Variables

The sociodemographic and clinical data were collected by reviewing the clinical histories. The sociodemographic and clinical variables collected were sex, family history of cancer and CRC, smoking status, drinking status, body mass index (BMI), type of CRC, age of cancer diagnosis, ECOG score, tumor size, histopathology, lymph node involvement, histological grade, metastasis, stage, TNM stage, primary tumor resection, and adjuvant chemotherapy. The patients were classified as smokers if they had smoked or were smoking 100 or more cigarettes in the course of their lives, ex-smokers if they had smoked 100 or more cigarettes in their lives but did not currently smoke, and nonsmokers if they had never smoked or had smoked less than 100 cigarettes in their lives. Individuals were classified by standard drink units (SDUs) as nondrinkers if they were teetotalers or did not consume alcohol regularly, as drinkers if their alcohol consumption was greater than 4 SDUs per day in men and greater than 2.5 SDUs in women, and as ex-drinkers if they did not drink currently but their alcohol consumption had been greater than 4 SDUs per day in men and greater than 2.5 SDUs in women [19]. Information on tumor histology and stage (tumor size, lymph node involvement, metastasis), whether the primary tumor had been resected, and the adjuvant chemotherapy treatment was also collected. The AJCC guide to staging system criteria was used for tumor classification [20].

### 2.5. Genetic Variables

#### 2.5.1. DNA Isolation

Following the patients’ inclusion and signing of the informed consent, saliva samples were collected with buccal swabs (OCR-100 Kit). The DNA was extracted using the QIAamp DNA Mini extraction kit (Qiagen GmbH, Hilden, Germany), following the specifications provided by the manufacturer for purifying DNA from saliva, and stored at −40 °C. The DNA concentration and purity were measured using a NanoDrop 2000 UV spectrophotometer with the absorbance ratio at 280/260 and 280/230.

#### 2.5.2. Detection of Gene Polymorphisms

The polymorphisms were analyzed by real-time polymerase chain reaction (PCR) using TaqMan probes (ABI Applied Biosystems, 7300 Real-Time PCR System). The polymorphisms analyzed and their assay IDs used are shown in Table 1. The genetic variables were determined with QuantStudio 12K Flex software (96 wells). The determination of the genetic variants was performed between the Pharmacogenetics Unit of the HUVN and the Department of Biochemistry and Molecular Biology II of the University of Granada. The criteria established for quality control of the SNPs were (1) missing genotype rate per SNP < 0.05; (2) minor allele frequency (MAF) > 0.01; (3) *p*-value in Hardy-Weinberg equilibrium test > 0.05; (4) missing genotype rate between cases and controls < 0.05. Variants included in the study were selected according to their frequency in the Caucasian population and previous results of similar studies in the scientific literature. 

### 2.6. Survival Variables

Progression-free survival (PFS) and overall survival (OS) were the variables used for analysis of the patients’ survival. Overall survival was defined as the time that elapsed from cancer diagnosis to death or last follow-up. Progression-free survival was calculated as the time from the start of therapy (for patients who received adjuvant therapy) or the time from the primary tumor resection (for subjects who did not receive adjuvant therapy) to last follow-up, death, or progression. The data on mortality were collected from the clinical histories.

### 2.7. Statistical Analysis

The descriptive analysis was performed using R 4.2.2 software [21]. The quantitative variables were expressed as the mean (±standard deviation) for those that complied with normality and as the median and percentiles (25 and 75) for the variables that did not follow a normal distribution. Normality was confirmed with the Shapiro–Wilk test.

For the analysis of association between survival and the sociodemographic, clinical, and genetic variables, we used the Kaplan–Meier method and a log-rank test. To perform the multivariate analysis, we used the Cox regression model (backward elimination method) and obtained the adjusted hazard ratio (HR) and the 95% confidence interval (95% CI) for the possible survival prognosis factors. All the tests were 2-tailed, and a probability of 0.05 or less was considered statistically significant.

The Hardy–Weinberg equilibrium and the MAFs were determined, and Lewontin’s D prime (D′) and the linkage disequilibrium coefficient (R^2^) were calculated. The linkage disequilibrium (LD) for each polymorphism was calculated with the PLINK genome association analysis program and Haploview 4.2 [22,23]. Haplotype survival analysis was performed using Thesias software 3.1, which applied Cox regression analysis [24]. 

## 3. Results

### 3.1. Patient Characteristics

The study included 127 patients from southern Spain diagnosed with colorectal cancer. The patients’ sociodemographic, clinical, and physio-pathological characteristics are shown in Table 2. The median follow-up was 35.8 (17.9–94.9) months for all CRC patients. Out of all the patients, 66.93% (85/127) were men; the mean age of CRC diagnosis was 62.52 ± 10.55 years, 44.09% (56/127) were nonsmokers, and 78.74% (100/127) had a BMI greater than 24. A total of 70.08% (89/127) of the patients were diagnosed with colon cancer, 55.91% (71/127) with stage I, II, or IIIA, and with surgery as first-line treatment in 93.70% of cases (119/127). Median OS and PFS for all patients were 82.3 (62.1-NR) (Appendix A) and 73.3 (41.0-NR) (Appendix A) months, respectively (Table 2).

### 3.2. Influence of Clinical-Pathological Characteristics on Survival

#### 3.2.1. Overall Survival

Median OS was higher in patients with a family history of CRC (*p*_log-rank_ = 0.020, 134.5 vs. 62.1 months; Appendix A), with a lower age of CRC diagnosis (*p*_log-rank_ < 0.001; Appendix A), and an ECOG score of 0–1 (*p*_log-rank_ = 0.001, 114.9 vs. 36.3 months; Appendix A), without metastasis (*p*_log-rank_ = 0.001, 134.5 vs. 56.2 months; Appendix A), with stage I, II, or IIIA (*p*_log-rank_ = 0.001, 134.5 vs. 52.1 months; Appendix A), fewer lymph nodes affected (*p*_log-rank_ = 0.001, 108.5 months for 1–3 vs. 30.6 months for >4; Appendix A), and with adjuvant fluoropyrimidine and oxaliplatin chemotherapy (*p*_log-rank_ = 0.003, 134.5 vs. 41.5 months for fluoropyrimidine + biologic, 41.9 months for fluoropyrimidine, and 30.2 months without adjuvant chemotherapy; Appendix A).

#### 3.2.2. Progression-Free Survival

A higher PFS value was found in patients with a lower age of CRC diagnosis (*p*_log-rank_ = 0.001; Appendix A), ECOG score 0–1 (*p*_log-rank_ < 0.001, 105.8 vs. 18.6 months; Appendix A), no lymph node involvement (*p*_log-rank_ = 0.020, NR vs. 41 months; Appendix A), absence of metastasis (*p*_log-rank_ < 0.001, 125.9 vs. 19.8 months; Appendix A), in stages I, II, or IIIA (*p*_log-rank_ < 0.001, 125 vs. 35 months; Appendix A), with fewer lymph nodes affected (*p*_log-rank_ < 0.001, 63.3 months for 1–3 vs. 18.6 months for <4; Appendix A), primary tumor surgically resected (*p*_log-rank_ = 0.008, 105.2 vs. 14.1 months; Appendix A), and with adjuvant fluoropyrimidine and oxaliplatin chemotherapy (*p*_log-rank_ < 0.001, 125.4 vs. 38.1 months for fluoropyrimidine and 23.7 months for fluoropyrimidine + biologic; Appendix A). Furthermore, we found a tendency toward association between higher median PFS and a history of CRC (*p*_log-rank_ = 0.080, 125.9 vs. 55.1 months for subjects with no history of CRC; Appendix A). 

### 3.3. Genotypes and Haplotypes Distribution

All the polymorphisms show MAFs greater than 1%, and none of them were excluded from the analysis (Appendix A). The values of the expected genotype frequencies, as well as the values of the Hardy–Weinberg equilibrium model for OS and PFS, are shown in Appendix A, respectively. The SNPs that did not comply with the Hardy–Weinberg equilibrium for OS and PFS were *GC* rs7041 (OS: *p* = 0.039; PFS: *p* = 0.007), *VDR* rs1544410 (OS: *p* = 0.006; PFS: *p* = 0.007), *VDR* rs2228570 (PFS: *p* = 0.034), *VDR* rs7975232 (OS: *p* = 0.047; PFS: *p* = 0.047), *VDR* rs731236 (OS: *p* = 0.016; PFS: *p* = 0.017), and *CYP24A1* rs6068816 (*p* = 0.003; PFS: *p* = 0.003). No significant differences were found between the MAFs described for the Iberian population and the frequencies obtained in our population for the SNPs studied: *GC* rs7041-A: 0.492 vs. 0.435 (*p* = 0.965), *VDR* rs1544410-T: 0.445 vs. 0.439 (*p* = 0.993), *VDR* rs2228570-A: 0.343 vs. 0.327 (*p* = 0.981), *VDR* rs7975232-C: 0.421 vs. 0.430 (*p* = 0.989), *VDR* rs731236-G: 0.402 vs. 0.430 (*p* = 0.968), *CYP24A1* rs6068816-T: 0.189 vs. 0.107 (*p* = 0.870) [6]. The linkage disequilibrium values (D’ and R2) are shown in Appendix A. Specifically, the SNPs *VDR* rs1544410/rs7975232, *VDR* rs1544410/rs731236, *VDR* rs7975232/rs731236, *CYP27B1* rs3782130/rs4646536, *CYP27B1* rs3782130/rs703842, *CYP27B1* rs3782130/rs10877012, *CYP27B1* rs4646536/rs703842, *CYP27B1* rs4646536/rs10877012, *CYP27B1* rs703842/rs10877012 are in strong linkage disequilibrium (Figure 1). The SNP rs1544410, rs7975232, and rs731236 were included in *VDR* haplotype block, and rs3782130, rs4646536, rs703842, and rs10877012 were included in *CYP27B1* haplotype. Haplotype frequency estimation values are given in Appendix A. The most frequent *VDR* haplotype was TAG (total frequency = 0.3766), and for *CYP27B1* block GAAG was the more frequent haplotype (total frequency = 0.7165). 

### 3.4. Influence of Gene Polymorphisms and Haplotypes on Survival

#### 3.4.1. Overall Survival

The bivariate analysis showed an association between OS and the *VDR* rs7975232 (ApaI) polymorphism. Specifically, patients carrying the *VDR* rs7975232 (ApaI)-AA genotype showed lower survival when compared to carriers of the *VDR* rs7975232 (ApaI)-C allele (*p* = 0.037, HR = 1.187, 95% CI = 1.039–3.377; Appendix A). The Kaplan–Meier graph with the OS curves for the *VDR* rs7975232 (ApaI)-AA vs. C genotype is shown in Figure 2 (*p*_log-rank_ = 0.030). Median OS was 134.5 months (95% CI = 69.3–NR) for the *VDR* rs7975232 (ApaI)-C allele compared to 48.9 months (95% CI = 39.6–NR) for patients carrying the *VDR* rs7975232 (ApaI)-C genotype. Similarly, a tendency was found toward association between OS and the *VDR* rs7975232 (ApaI)-AA vs. CC/AC genotypes (*p* = 0.062, HR = 2.181, 95% CI = 0.962–4.944; Appendix A). The Kaplan–Meier graph with the OS curves for the *VDR* rs7975232 (ApaI)-AA vs. CC/AC genotypes is shown in Figure 3 (*p*_log-rank_ = 0.090). Median OS was 114.9 months (95% CI = 61.3–NR) for the *VDR* rs7975232 (ApaI)-AC genotype, compared to 48.9 months (95% CI = 39.6–NR) for patients with the *VDR* rs7975232 (ApaI)-AA genotype. Along the same lines, we found a tendency toward association between the *VDR* rs731236 (TaqI) SNP and OS. Patients with the *VDR* rs731236 (TaqI)-GG genotype showed lower survival than carriers of the *VDR* rs731236 (TaqI)-AG genotype (*p* = 0.034, HR = 2.409, 95% CI = 0.818–3.271; Appendix A). The Kaplan–Meier graph with the OS curves for the *VDR* rs731236 (TaqI)-GG vs. AG/AA genotype is shown in Figure 4 (*p*_log-rank_ = 0.090). Median OS was 134.5 months (95% CI = 62.3–NR) for the *VDR* rs731236 (TaqI)-AG genotype compared to 41.9 months (95% CI = 36.3–NR) and 75.9 months (95% CI = 58.5–NR) for patients carrying *VDR* rs731236 (TaqI)-GG and AA, respectively. Furthermore, the same tendency toward an association was found when the *VDR* rs731236 (TaqI)-A allele was compared to the *VDR* rs731236 (TaqI)-GG genotype (GG vs. A: *p* = 0.087, HR = 1.825, 95% CI = 0.915–3.637; Appendix A). The Kaplan–Meier graph of the OS curves for the *VDR* rs731236 (TaqI)-GG vs. A genotype is shown in Figure 5 (*p*_log-rank_ = 0.080). Median OS was 134.5 months (95% CI = 62.3–NR) for the *VDR* rs731236 (TaqI)-A allele compared to 41.9 months (95% CI = 36.3–NR) for patients carrying *VDR* rs731236 (TaqI)-GG. Moreover, we found a tendency toward association between the *CYP24A1* rs6068816 SNP and the risk of lower OS. Specifically, patients with the *CYP24A1* rs6068816-TT genotype showed lower survival than carriers of the *CYP24A1* rs6068816-C allele (*p* = 0.067, HR = 2.244, 95% CI = 0.946–5.327; Appendix A). Figure 6 shows the Kaplan–Meier graph of the OS curves for this association tendency (*p*_log-rank_ = 0.060). Median OS was 108.5 months (95% CI = 62.3–NR) for the *CYP24A1* rs6068816-C allele compared to 61.3 months (95% CI = 23.7–NR) for the *CYP24A1* rs6068816-TT genotype. Finally, the *GC* rs7041 SNP showed an impact on OS in our patients. For example, individuals with the *GC* rs7041-C allele show lower OS than carriers of the *GC* rs7041-AA genotype (*p* = 0.020, HR = 2.914, 95% CI = 1.149–7.393; Appendix A). The Kaplan–Meier graph of the OS curves is shown in Figure 7 (*p*_log-rank_ = 0.020). Median OS was 65.5 months (95% CI = 53.5–NR) for subjects with the *GC* rs7041-C allele, compared to 69.3 months (95% CI = 56.2–NR) for those carrying the *GC* rs7041-A allele. In addition, when we studied the association between OS and the *GC* rs7041-AC vs. AA/CC genotypes, we found that individuals carrying the *GC* rs7041-AC genotype showed lower survival (*p* = 0.009, HR = 3.514, 95% CI = 1.359–9.088; Appendix A) than carriers of the *GC* rs7041-CC and AA genotypes. The Kaplan–Meier graph of the OS curves is shown in Figure 8 (*p*_log-rank_ = 0.010). Median OS was 61.3 months (95% CI = 41.9–NR) for subjects with the *GC* rs7041-AC genotype, compared to 114.5 months (95% CI = 65.0–NR) for those carrying the *GC* rs7041-CC genotype. Regarding the overall survival haplotype analysis, the frequencies of *VDR* and *CYP27B1* haplotypes in censored and non-censored patients are shown in Appendix A, respectively. Likewise, the results of Cox regression analysis for haplotypes and overall survival revealed no significant findings related to *VDR* block (Appendix A) (al *p*-value > 0.05). As for the *CYP27B1* haplotype analysis, only an association analysis could be performed due to its low frequency in our patients. This analysis showed no significant association between the CGGT haplotype and overall survival in our patients (*p* = 0.473, HRR = 0.84, 95% CI = 0.54–1.33, CGGG vs. CGGT). The Cox regression analysis adjusted for no family history of CRC, stage (IIIB, IV, or IVB), adjuvant chemotherapy, lymph node involvement, ECOG score (2–4), metastasis, and age of diagnosis showed that the *CYP24A1* rs6068816-TT (*p* = < 0.001, HR = 6.237, 95% CI = 2.297–16.936; Table 3) and *GC* rs7041-C (*p* = 0.006, HR = 4.939, 95% CI = 1.578–15.457; Table 3) and *VDR* rs7975232 (ApaI)-AA (*p* = 0.036, HR = 1.974, 95% CI = 1.045–3.731; Table 3) SNPs were the independent variables associated with OS in patients diagnosed with CRC (*p_log-rank_* < 0.001) (Table 3). 

#### 3.4.2. Progression-Free Survival

Three of the five SNPs of the *VDR* gene analyzed showed an association with PFS after the bivariate analysis was performed. First, patients carrying the *VDR* rs2228570 (FokI)-GG and AA genotypes showed a higher risk of progression compared to carriers of the *VDR* rs2228570 (FokI)-AG genotype (GG: *p* = 0.085, HR = 1.650, 95% CI = 0.934–2.916; AA: *p* = 0.025, HR = 2.542, 95% CI = 1.126–5.742; Appendix A). Figure 9 shows the Kaplan–Meier graph with the PFS curves (*p*_log-rank_ = 0.050). Median survival was 125.4 months (95% CI = 47.4–NR) for patients carrying the VDR rs2228570 (FokI)-AG genotype, compared to 55.1 months (95% CI = 23.7–NR) and 19.8 months (95% CI = 6.23–NR) for patients carrying the *VDR* rs2228570 (FokI)-GG and AA genotypes, respectively. Similarly, a tendency was found toward an association between the *VDR* rs2228570 (FokI)-AA genotype and higher risk of progression, when compared to the *VDR* rs2228570 (FokI)-G allele (*p* = 0.072, HR = 1.994, 95% CI = 0.939–4.232; Appendix A). The Kaplan–Meier graph with the PFS curves for the *VDR* rs2228570 (FokI)-AA vs. G allele is shown in Figure 10 (*p*_log-rank_ = 0.070). Median PFS for patients carrying the *VDR* rs2228570 (FokI)-G allele was 75.0 months (95% CI = 43.8–NR), compared to 19.8 months (95% CI = 6.23–NR) for patients carrying the *VDR* rs2228570 (FokI)-AA genotype. Second, the *VDR* rs7975232 (ApaI) polymorphism was also found to be associated with PFS. The *VDR* rs7975232 (ApaI)-AA genotype showed a higher risk of progression than the VDR rs7975232 (ApaI)-C allele (*p* = 0.042, HR = 1.172, 95% CI = 1.019–2.945; Appendix A). Figure 11 shows the Kaplan–Meier graph with the PFS curves for this association (*p*_log-rank_ = 0.040). Patients carrying the *VDR* rs7975232 (ApaI)-AA genotype had a median PFS of 43.8 months (95% CI = 34.5–NR), compared to 125.9 months (95% CI = 55.1–NR) for carriers of the *VDR* rs7975232 (ApaI)-C allele. Finally, we found a tendency toward association between the *VDR* rs731236 (TaqI)-GG genotype and higher risk of progression in our patients, compared to the *VDR* rs731236 (TaqI)-A allele (*p* = 0.094, HR = 1.670, 95% CI = 0.916–3.046; Appendix A). The Kaplan–Meier graph with the PFS curves for this SNP is shown in Figure 12 (*p*_log-rank_ = 0.090). Median PFS for subjects carrying the *VDR* rs731236 (TaqI)-GG genotype was 41.0 months (95% CI = 18.6–NR), compared to 105.0 months (95% CI = 51.1–NR) for carriers of the *VDR* rs731236 (TaqI)-A allele. Moreover, the *CYP24A1* rs6068816-TT genotype showed a higher risk of progression when compared to the *CYP24A1* rs6068816-C allele (*p* = 0.006, HR = 3.030, 95% CI = 1.361–6.743; Appendix A). The Kaplan–Meier graph with the PFS curves for this association is shown in Figure 13 (*p*_log-rank_ = 0.004). Median PFS for subjects with the *CYP24A1* rs6068816-C allele was 75 months (95% CI = 47.4–NR), compared to 14.2 months (95% CI = 3.1–NR) for patients carrying the *CYP24A1* rs6068816-TT genotype. Furthermore, when the *CYP24A1* rs6068816-TT vs. CC/CT genotypes were compared, it was found that patients carrying the *CYP24A1* rs6068816-TT genotype showed a higher risk of progression (*p* = 0.010, HR = 3.585, 95% CI = 1.355–6.743; Appendix A) than those with the *CYP24A1* rs6068816-CC and CT genotypes. The Kaplan–Meier graph with the PFS curves for this association is shown in Figure 14 (*p*_log-rank_ = 0.010). Median PFS for subjects with the *CYP24A1* rs6068816-CC genotype was 63.3 months (95% CI = 43.8–NR), compared to 14.2 months (95% CI = 3.1–NR) for patients carrying the *CYP24A1* rs6068816-TT genotype. Similarly, individuals with the *GC* rs7041-AC and AA genotypes showed a higher risk of progression when compared to carriers of the *GC* rs7041-CC genotype (AA: *p* = 0.777, HR = 1.149, 95% CI = 0.440–2.997; AC: *p* = 0.007, HR = 2.742; 95% CI = 1.321–5.691; Appendix A). The Kaplan–Meier graph with the PFS curves for this SNP is shown in Figure 15 (*p*_log-rank_ = 0.004). Median PFS for patients with the *GC* rs7041-CC genotype was 125.9 months (95% CI = 105.8–NR) compared to 55.1 months (95% CI = 37.0–NR) and 38.1 months (95% CI = 18.5–73.3) for carriers of the *GC* rs7041-AA and AC genotypes, respectively. Furthermore, when the *GC* rs7041-A allele was analyzed in comparison to the *GC* rs7041-CC genotype, a higher risk of progression was found in subjects carrying the *GC* rs7041-A allele (*p* = 0.029, HR = 2.226, 95% CI = 1.083–4.574; Appendix A). The Kaplan–Meier graph with the PFS curves for this association is shown in Figure 16 (*p*_log-rank_ = 0.030). Median PFS for patients carrying the *GC* rs7041-CC genotype was 125.9 months (95% CI = 105.8–NR) compared to 55.1 months (95% CI = 37.0–NR) for carriers of the *GC* rs7041-A allele. The frequencies of *VDR* and *CYP27B1* haplotypes in censored and non-censored patients are shown in Appendix A, respectively, for haplotype analysis of progression-free survival. Also, the results of Cox regression analysis for haplotypes and progression-free survival showed no significant results in relation to *VDR* (Appendix A) (all *p*-value > 0.05). As for the *CYP27B1* haplotype analysis, only an association analysis could be performed due to its low frequency in our patients. This analysis showed no significant association between the CGGT haplotype and overall survival in our patients (*p* = 0.927, HRR = 0.98, 95% CI = 0.67–1.44, CGGG vs. CGGT). The Cox regression analysis of PFS adjusted for age of diagnosis, ECOG score (2–4), lymph node involvement, metastasis, stage (IIIB, IV, or IVB), no primary tumor resection and adjuvant chemotherapy showed that the *CYP24A1* rs6068816-TT (*p* < 0.001, HR = 6.095, 95% CI = 2.413–15.388; Table 4) is associated and *VDR* rs7975232 (ApaI)-AA (*p* = 0.066, HR = 1.692, 95% CI = 1.402–2.092; Table 4) has a tendency to PFS in our patients (*p_log-rank_* < 0.001) (Table 4). 

## 4. Discussion

Various studies have highlighted the possible association of the polymorphisms involved in the vitamin D metabolic pathway with CRC survival, through influencing the disease progression mechanism [15,25,26,27]. One of the molecules that contribute to vitamin D metabolism is *VDR*, which participates in cell interaction and gene regulation processes [28,29,30]. Currently, several SNPs in the *VDR* gene that could be related to CRC have been identified [31]. Our study included 127 Caucasian patients (from Spain) diagnosed with CRC. After performing the multivariate analysis, we found a significant association between patients carrying the mutated *VDR* rs7975232 (ApaI)-AA genotype and lower OS and PFS compared to carriers of the rs7975232 (ApaI)-C allele. In line with our results, we found a meta-analysis conducted in 1588 patients of various ethnicities (from the United States, Taiwan, and China) diagnosed with cancer, in which the *VDR* rs7975232 (ApaI)-AA genotype was associated with lower overall survival (*p* = 0.0001, HR = 1.77, 95% CI = 0.79–2.75, I2 = 0.95; AA vs. CC), as well as with an increased risk of progression of the disease in patients (*p* < 0.05, HR = 1.29, 95% CI = 1.02–1.56, I2 = 0; AA vs. CC) [14]. For the *VDR* rs731236 (TaqI) SNP, our results showed a tendency toward an association between carriers of the *VDR* rs731236 (TaqI)-GG genotype and lower OS and PFS compared to those carrying the wild-type *VDR* rs731236 (TaqI)-A allele. A study conducted in 1759 Caucasian patients (from Germany) diagnosed with breast cancer showed that the mutated homozygous genotype *VDR* rs731236 (TaqI)-GG had three times more risk of death from the disease than the *VDR* rs731236 (TaqI)-AA genotype in the study cohort (*p*_trend_ = 0.023, HR = 2.80, 95% CI = 1.10–7.20; GG vs. AA) [32]. In contrast to our results, a study carried out in 893 Caucasian patients (from Italy) diagnosed with metastatic CRC under treatment with various chemotherapy regimens (FOLFIRI + cetuximab [n = 278]/bevacizumab [n = 522]/irinotecan ± cetuximab [n = 93]) showed that for the FOLFIRI + bevacizumab cohort, OS was lower in patients carrying the *VDR* rs731236 (TaqI)-AA/AG genotypes (23.7 vs. 28.8 months, *p* = 0.046, HR = 0.62, 95% CI = 0.39–0.99; GG vs. AA/AG) [33]. Another of the *VDR* SNPs that have been investigated is *VDR* rs2228570 (FokI). Specifically, the wild-type *VDR* rs2228570 (FokI)-AA genotype was found to be associated with lower PFS when compared to the *VDR* rs2228570 (FokI)-GG and AG genotypes. A study conducted by Fedirko et al. in 1095 Caucasian patients (from Europe) diagnosed with CRC found no significant association between this SNP and cancer survival in the patients analyzed (*p* = 0.270, HR = 0.81, 95% CI = 0.60–1.10) [34]. Another study performed in Caucasian patients (294 cases) (from the United States) diagnosed with advanced non-small-cell lung cancer (NSCLC) showed that carriers of the *VDR* rs2228570 (FokI)-AA and AG genotypes had a higher risk of death (*p* = 0.040, HR = 1.32, 95% CI = 0.98–1.77; AG vs. GG and *p* = 0.04, HR = 1.41, 95% CI = 0.96–2.07; AA vs. GG) [35]. As for the *VDR* rs11568820 (Cdx2) SNP, a clinical trial in 155 Asian patients (from Japan) diagnosed with NSCLC showed that individuals carrying the *VDR* rs11568820 (Cdx2)-TT/CT genotypes showed higher OS after 5 years (*p* = 0.040, HR = 0.41, 95% CI = 0.18–0.97; TT/CT vs. CC) [36]. In contrast to these results, another study in 194 Caucasian patients (from Spain) diagnosed with NSCLC associated the *VDR* rs11568820 (Cdx2)-TT genotype with lower overall survival in the patients (*p* = 0.013, HR = 7.43, 95% CI =1.53–36.15; TT vs. CC/CT) [25]. In our study, however, we found no association between this polymorphism and the survival of the patients included in the cohort. The last SNP included in this study of the *VDR* gene is *VDR* rs1544410 (BsmI). A meta-analysis that included 44,165 Caucasian patients (from the United Kingdom), from 64 studies, diagnosed with cancer, associated the mutated genotypes *VDR* rs1544410 (BsmI)-TT and CT with lower patient survival compared to the wild-type *VDR* rs1544410 (BsmI)-CC genotype (*p* < 0.050, HR=1.40, 95% CI = 1.05–1.75, I2 = 0.85; TT/CT vs. CC) [14]. In our study, we found no significant results between the *VDR* rs1544410 (BsmI) SNP and OS and PFS in CRC. The results obtained after analysis of the SNPs located in *VDR* show an association between the altered alleles and lower survival in CRC. These polymorphisms could give rise to a modified receptor, affecting its interaction with vitamin D [37]. Both vitamin D metabolism and the actions of the *VDR* would therefore be reduced, which could favor tumor progression [37,38,39].

The *CYP24A1* gene, located on the long arm of chromosome 20 (20q13.2), encodes the 24-hydroxylase enzyme, responsible for catalyzing calcitriol in its inactive forms [40]. There are currently few studies that examine the relationship between the *CYP24A1* rs6068816 and rs4809957 SNPs and CRC, with contradictory results [41,42]. As regards the *CYP24A1* rs6068816 SNP, our study has shown an association between the mutated genotype *CYP24A1* rs6068816-TT and low OS and PFS values after the multivariate analysis, compared to the *CYP24A1* rs6068816-C allele. Along the same lines, a study carried out in 146 Caucasian patients (from Spain) diagnosed with NSCLC associated the *CYP24A1* rs6068816-TT genotype with lower survival (*p* = 0.005, HR = 3.75, 95% CI = 1.49–9.41; TT vs. C) and greater progression of the disease in non-resected patients (*p* = 0.018, HR = 2.99, 95% CI = 1.21–7.45; TT vs. C) [25]. On the other hand, Gong et al. showed that the wild-type AA genotype of the *CYP24A1* rs4809957 SNP conferred a worse CRC prognosis in an Asian population (524 cases/595 controls) (*p* < 0.050, HR = 2.38, 95% CI = 1.30–4.37; AA vs. GG) [43]. However, in our study we found no significant association between *CYP24A1* rs4809957 and OS and PFS. The *CYP24A1* rs4809957 and rs6068816 SNPs, located in intronic regions, could affect the mRNA splicing process in transcription and translation [43]. Specifically, these SNPs have been related to increased expression of CYP24A, and therefore a larger quantity of enzyme [44]. This would result in an increase in enzyme activity and greater metabolization of calcitriol, reducing its plasma levels as well as its antineoplastic activity [41]. Patients carrying mutated alleles could therefore have a worse prognosis in respect of CRC survival [41,44,45].

Another of the enzymes involved in the vitamin D metabolic pathway is CYP27B1, which catalyzes the conversion of 25-hydroxyvitamin D into its active form (1,25-dihydroxyvitamin D) in the liver [11]. In the gene responsible for encoding this enzyme, some polymorphisms have been found that could influence its synthesis and function. However, in our study we did not find any association between the SNPs studied in the *CYP27B1* gene and survival in CRC. In line with our results, a study conducted in 893 Caucasian patients (from Italy and the United States) diagnosed with metastatic CRC found no association between the *CYP27B1* rs703842 SNP and OS (*p* = 0.940) and PFS (*p* = 0.280) in its patients [33]. As for the *CYP27B1* rs3782130 SNP, this was previously studied in patients with NSCLC without finding any association with overall survival [25,46]. Gong et al. carried out a study in Asian patients (from China) diagnosed with CRC (528 cases/605 controls) in which they associated the wild-type *CYP27B1* rs4646536-AA genotype with a worse disease prognosis (*p*_log-rank_ = 0.010; AA vs. GG and AG) [43]. On *CYP27B1* rs10877012, a study conducted in 542 Asian patients (from China) diagnosed with NSCLC found no association between *CYP27B1* rs10877012 and OS (*p* = 0.695; GG/TG vs. TT) [46]. Some studies have associated greater expression of the *CYP27B1* gene, as well as greater histological differentiation of the tumor, with an increase in OS in patients diagnosed with lung cancer; however, this mechanism is not entirely clear [47,48].

One of the main metabolites in the vitamin D pathway is calcidiol, which is synthesized by 25-hydroxylase. This enzyme is encoded by the *CYP2R1* gene, located in the 11p15.2 region [49]. In our study, we did not find an association between the *CYP2R1* rs10741657 SNP and CRC survival. In line with our results, a study in 194 Caucasian patients (from Spain) diagnosed with NSCLC found no association when it investigated this SNP and OS (*p* > 0.050) [25]. Some studies have related the mutated genotype *CYP2R1* rs10741657-GG to alterations in vitamin D plasma levels and a worse cancer prognosis, due to malfunctioning of the 25-hydroxylase enzyme caused by this SNP [50,51].

The VDBP peptide sequence consists of 458 amino acids, arranged in three domains (I, II, and III), which can be glycosylated to a variable degree depending on various SNPs [52]. The *GC* gene, located on chromosome 4, responsible for synthesizing it, has a SNP, *GC* rs7041, located on exon 11 (which encodes domain III), that could be involved in the formation of the three main types of VDBP (GCF, GCS, and GC2) [52]. Our results show that individuals carrying the mutated allele *GC* rs7041-C have lower OS compared to those carrying the *GC* rs7041-AA genotype. Moreover, OS and PFS were lower in patients with the *GC* rs7041-AC genotype. In line with our results is the project carried out by Pibiri et al. in a Caucasian and African American population (from the United States) diagnosed with CRC (961 cases/838 controls), in which it was observed that the *GC* rs7041-C allele was associated with a worse CRC prognosis, affecting survival unfavorably (*p* = 0.059, HR = 0.77, 95% CI = 0.58–1.01; AA vs. C) [53]. The presence of the mutated allele *GC* rs7041-C could give rise to altered VDBP molecules or to a reduction in its synthesis and lower plasma levels [52,54]. Consequently, the transport of active vitamin D metabolites, and therefore vitamin D levels, could be impaired [52,54]. The disruption of this transport mechanism could be one of the causes of worse CRC prognosis in our patients [52,54,55].

This is the first study in which the *VDR* rs11568820 (Cdx2), *CYP24A1* rs6068816, and *CYP2R1* rs10741657 SNPs are related to survival in patients with CRC. One of its strengths is that it was conducted in a cohort of patients diagnosed with CRC at the same hospital, following the same protocols, ensuring the uniformity of the study population. However, the inconsistency of our results with the ones reported from some prior studies might be due to the number of patients included in the study cohorts and the different origins of the participants. Thus, future studies with larger cohorts are needed to establish and be able to detect other variants related to CRC survival. We can therefore state that we found an association between the SNPs involved in the mechanism of action of vitamin D and OS and PFS in CRC. Specifically, our results could suggest that the *VDR* rs7975232 (Apal), *VDR* rs731236 (TaqI), *VDR* rs2228570 (FokI), *CYP24A1* rs6068816, and *GC* rs7041 SNPs could act as predictors of survival in patients diagnosed with CRC. Validation of these SNPs in an independent cohort are needed to be used, in the future, as predictive biomarkers for the prognosis of the disease.

## 5. Conclusions

The *VDR* rs7975232 (ApaI)-AA as well as *GC* rs7041-C and *CYP24A1* rs6068816-TT polymorphisms could be a risk factor for lower OS as well as *CYP24A1* rs6068816-TT for PFS in Caucasian patients diagnosed with CRC; however, further research performed in higher populations is needed to demonstrate it. Moreover, patients carrying the *VDR* rs2228570 (FokI) SNP could have lower PFS. We did not find any relationship between the *CYP2R1* rs10741657, *CYP27B1* rs10877012, *CYP27B1* rs4646536, *CYP27B1* rs3782130, *CYP27B1* rs703842, *CYP24A1* rs4809957, *VDR* rs1544410 (BsmI), or *VDR* rs11568820 (Cdx2) SNPs and CRC survival. 

## Figures and Tables

**Figure 1 cancers-15-04077-f001:**
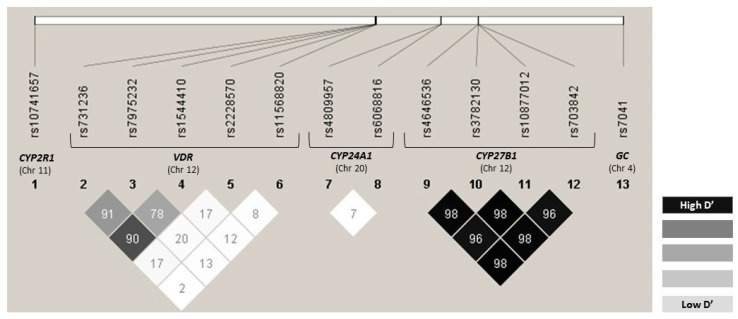
LD D’ plot for SNPs located in *VDR* gene, *CYP24A1* gene, and *CYP27B1* gene in the whole population. Numbers inside the squares are the D’ values expressed as a percent.

**Figure 2 cancers-15-04077-f002:**
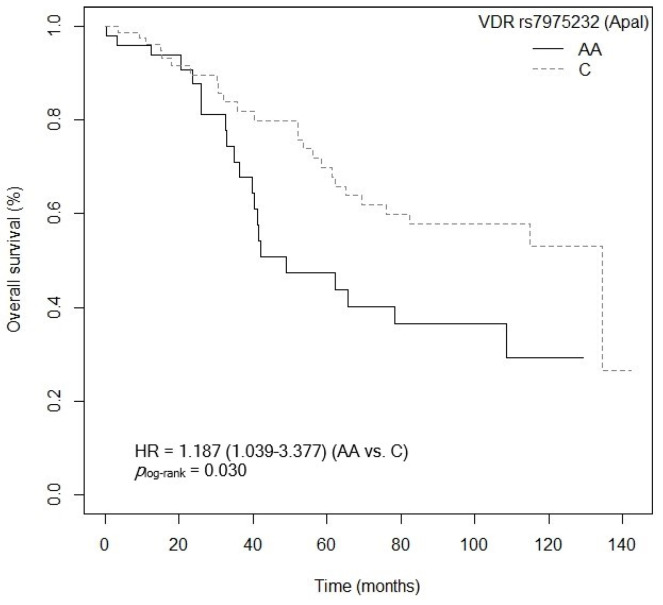
Kaplan–Meier plot of overall survival curves with the C allele of the *VDR* rs7975232 (ApaI) polymorphisms in 127 patients with CRC.

**Figure 3 cancers-15-04077-f003:**
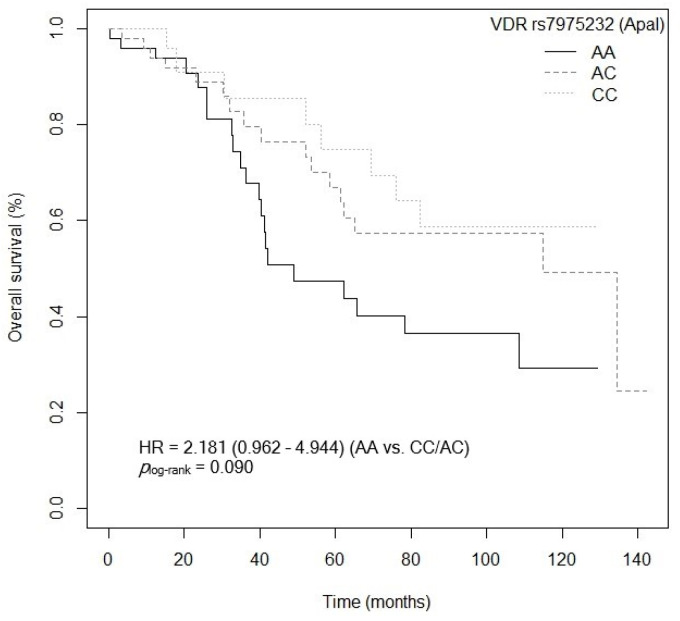
Kaplan–Meier plot of overall survival curves with the AA genotype of the *VDR* rs7975232 (ApaI) polymorphisms in 127 patients with CRC.

**Figure 4 cancers-15-04077-f004:**
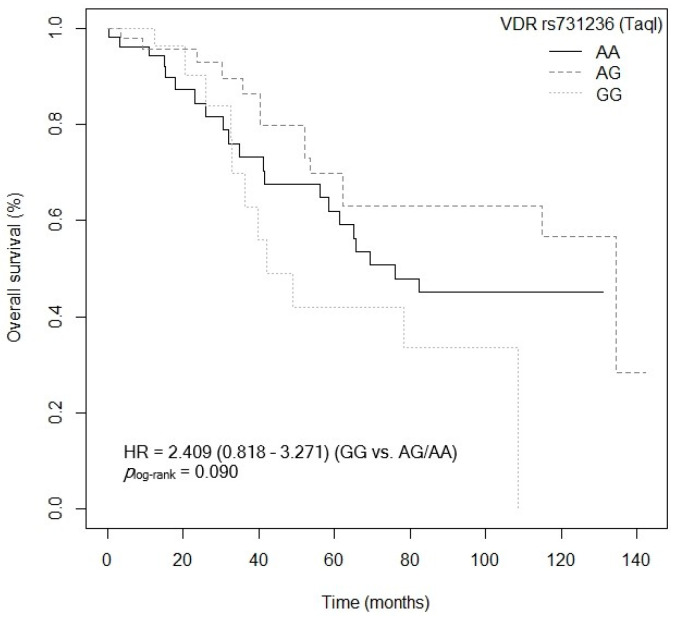
Kaplan–Meier plot of overall survival curves with the GG genotype of the *VDR* rs731236 (TaqI) polymorphisms in 127 patients with CRC.

**Figure 5 cancers-15-04077-f005:**
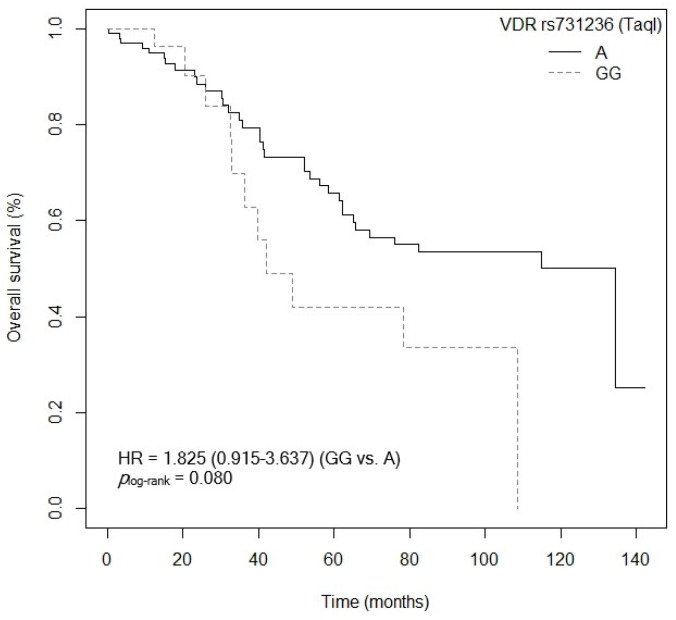
Kaplan–Meier plot of overall survival curves with the A allele of the *VDR* rs731236 (TaqI) polymorphisms in 127 patients with CRC.

**Figure 6 cancers-15-04077-f006:**
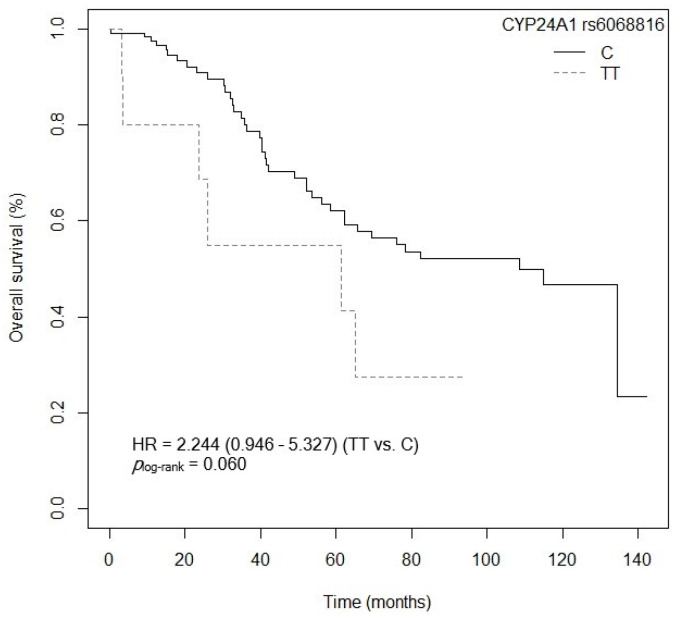
Kaplan–Meier plot of overall survival curves with the C allele of the *CYP24A1* rs6068816 polymorphisms in 127 patients with CRC.

**Figure 7 cancers-15-04077-f007:**
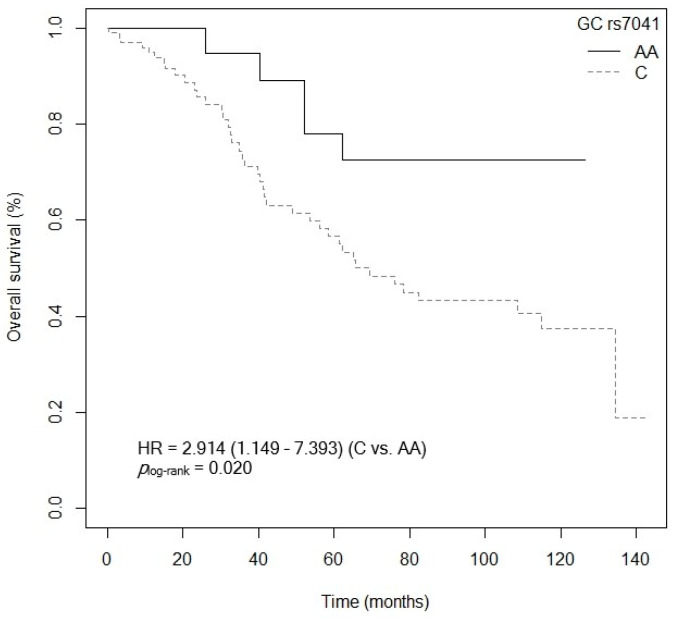
Kaplan–Meier plot of overall survival curves with the C allele of the *GC* rs7041 polymorphisms in 127 patients with CRC.

**Figure 8 cancers-15-04077-f008:**
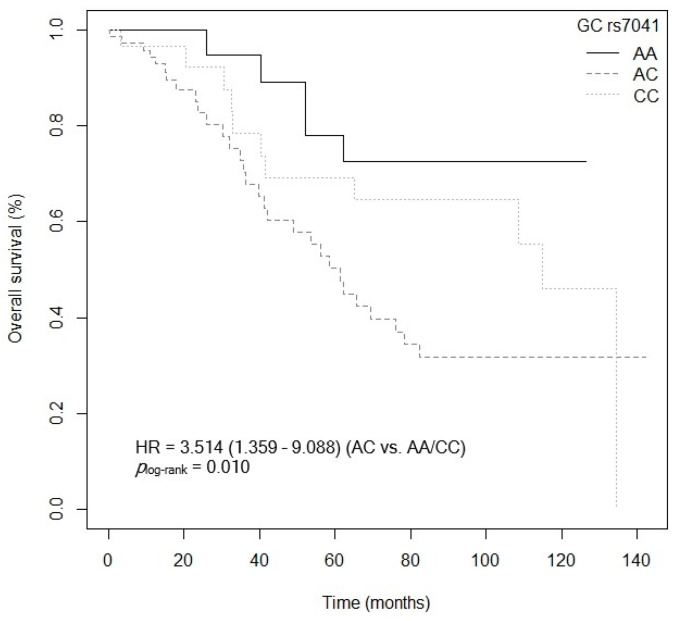
Kaplan–Meier plot of overall survival curves with the AC genotype of the *GC* rs7041 SNP in 127 patients with CRC.

**Figure 9 cancers-15-04077-f009:**
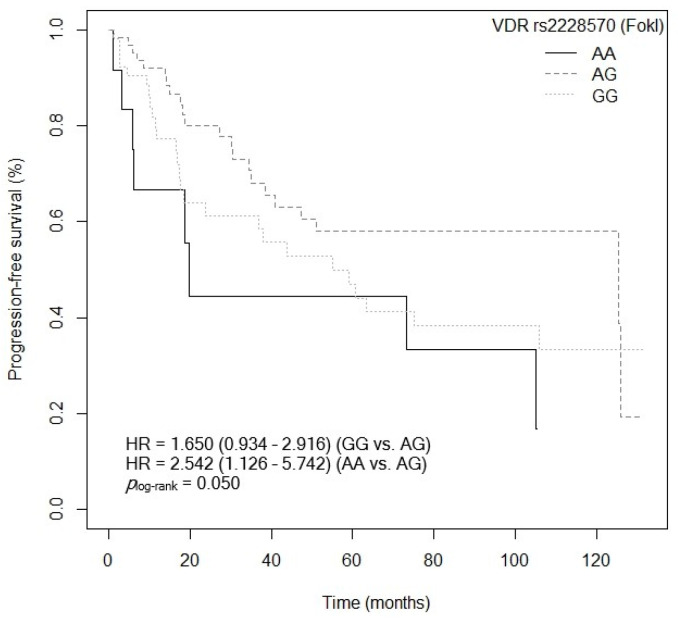
Kaplan–Meier plot of progression-free survival curves with the AG genotype if the *VDR* rs2228570 (FokI) polymorphism in 127 patients with CRC.

**Figure 10 cancers-15-04077-f010:**
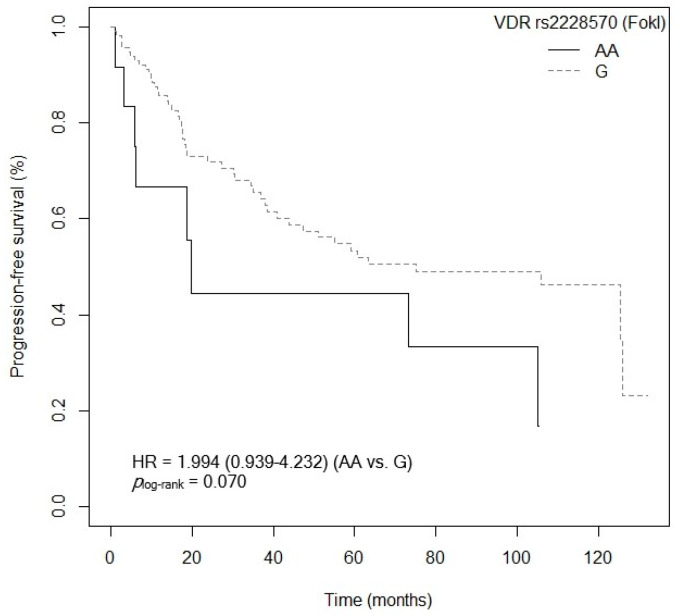
Kaplan–Meier plot of progression-free survival curves with the G allele of the *VDR* rs2228570 (FokI) polymorphism in 127 patients with CRC.

**Figure 11 cancers-15-04077-f011:**
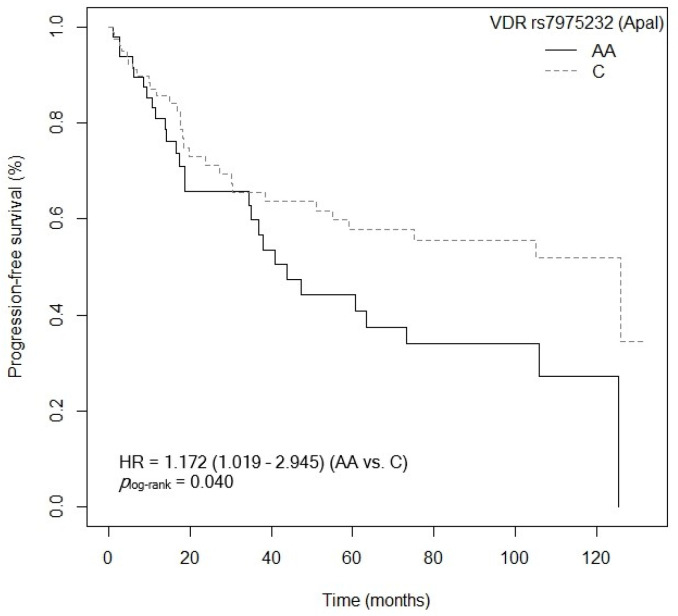
Kaplan–Meier plot of progression-free survival curves with the AA genotype of the *VDR* rs7975232 (ApaI) polymorphism in 127 patients with CRC.

**Figure 12 cancers-15-04077-f012:**
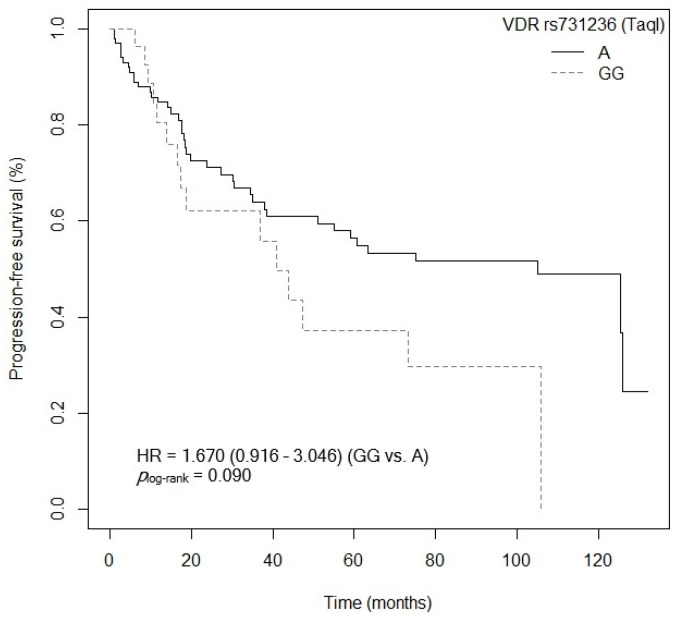
Kaplan–Meier plot of progression-free survival curves with the GG genotype of the *VDR* rs731236 (TaqI) polymorphism in 127 patients with CRC.

**Figure 13 cancers-15-04077-f013:**
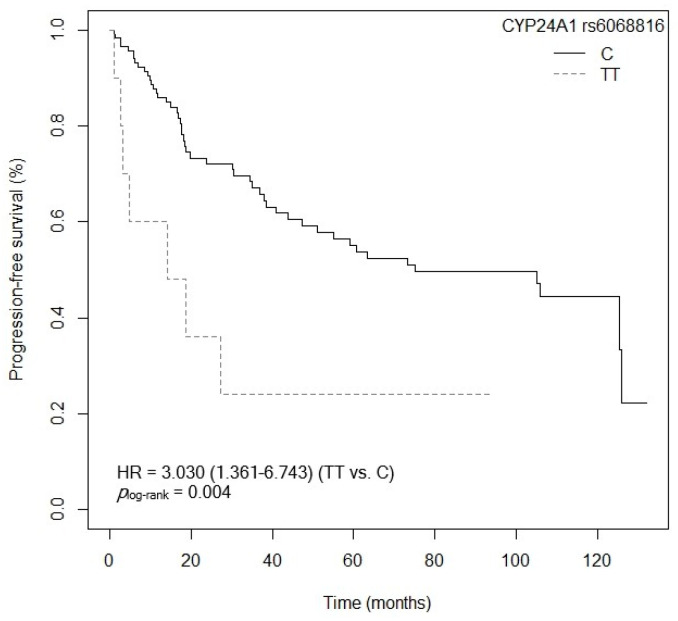
Kaplan–Meier plot of progression-free survival curves with the TT genotype in the dominant model of the *CYP24A1* rs6068816 polymorphism in 127 patients with CRC.

**Figure 14 cancers-15-04077-f014:**
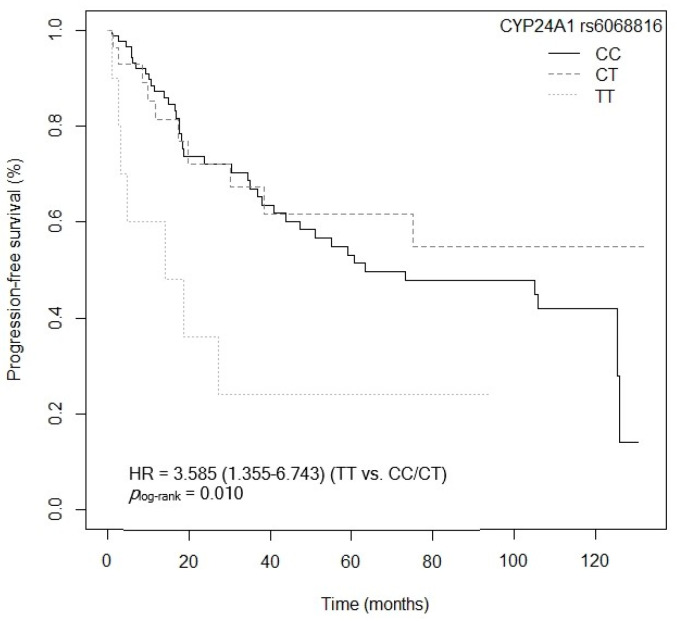
Kaplan–Meier plot of progression-free survival curves with the TT genotype in the genotypic model of the *CYP24A1* rs6068816 polymorphism in 127 patients with CRC.

**Figure 15 cancers-15-04077-f015:**
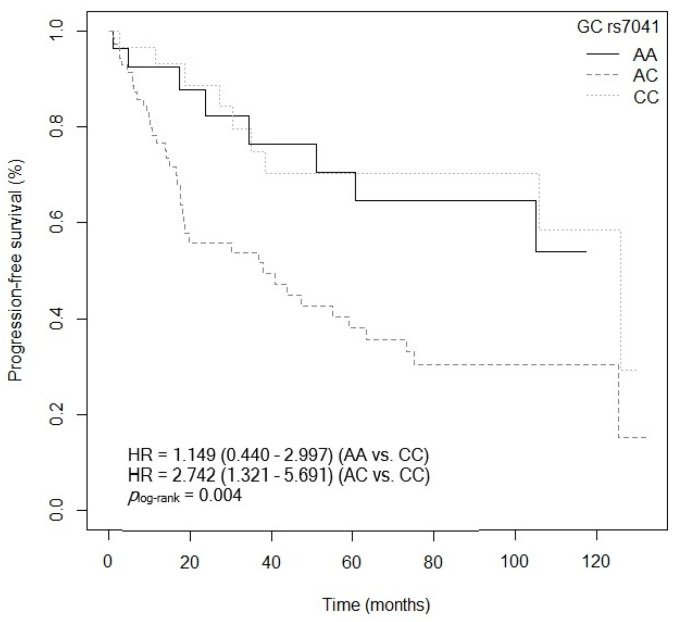
Kaplan–Meier plot of progression-free survival curves with the CC genotype of the *GC* rs7041 polymorphism in 127 patients with CRC.

**Figure 16 cancers-15-04077-f016:**
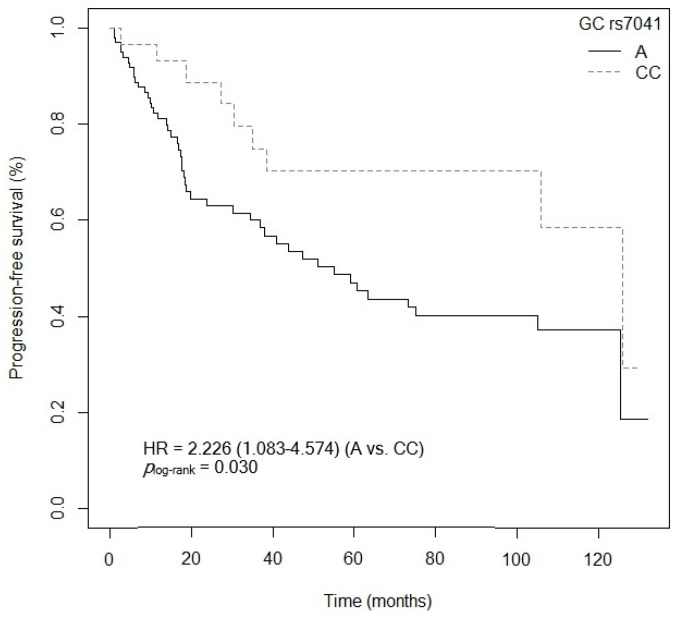
Kaplan–Meier plot of progression-free survival curves with the A allele of the *GC* rs7041 polymorphism in 127 patients with CRC.

**Table 1 cancers-15-04077-t001:** Gene SNPs and their TaqMan Assay IDs.

Gene	dbSNP ID	Location	Alleles	Assay ID
*VDR * (12q13.11)	rs1544410 (BsmI)	Intron 8	C > T	C___8716062_20
	rs11568820 (Cdx2)	Intron 1	C > T	C___2880808_10
	rs2228570 (FokI)	Exon 2	A > G	C__12060045_20
	rs7975232 (ApaI)	Intron 8	C > A	C__28977635_10
	rs731236 (TaqI)	Exon 9	A > G	C___2404008_10
*CYP27B1* (12q14.1)	rs4646536	Intron 6	A > G	C__25623453_10
	rs3782130	5′ Promoter	G > C	ANGZRHH ^1^
	rs10877012	5′ UTR	G > T	C__26237740_10
	rs703842	3′ UTR	A > G	AN9HX2K ^1^
*CYP24A1* (20q13.2)	rs6068816	Exon 6	C > T	C__25620091_20
	rs4809957	3′ UTR	A > G	C___3120981_20
*CYP2R1* (11p15.2)	rs10741657	5′ UTR	A > G	C___2958430_10
*GC* (4q13.3)	rs7041	Exon 11	A > C	C___3133594_30

^1^ The SNPs were analyzed using custom assays by ThermoFisher Scientific (Waltham, MA, USA). SNP: single nucleotide polymorphism; UTR: untranslated region.

**Table 2 cancers-15-04077-t002:** Sociodemographic, clinical, and physio-pathological characteristics of 127 CRC patients.

Variable	Baseline		
	N	%	Mean ± Standard Deviation
Sex	127	100	
Female	42	33.07	-
Male	85	66.93	-
Family history of cancer			
Yes	64	50.39	-
No	63	49.61	-
Family history of CRC			
Yes	27	21.26	-
No	100	78.74	-
Smoking status			
Smokers	27	21.26	-
Ex-smokers	44	34.65	-
Nonsmokers	56	44.09	-
Drinking status			
Drinkers	24	18.90	-
Ex-drinkers	7	5.51	-
Nondrinkers	96	75.59	-
Body mass index (BMI)			
<24	27	21.26	-
>24	100	78.74	-
Type of CRC			
Colon	89	70.08	-
Rectum	38	29.92	-
Age of diagnosis	127	-	62.52 ± 10.55
ECOG score			
0–1	116	91.34	-
2.4	11	8.66	-
Histopathology			
ADC	107	84.25	-
Mucinous ADC	20	15.75	-
Tumor size	127	-	4.40 (3.00–5.50)
Lymph node involvement			
Yes	92	72.44	-
No	35	27.56	-
Number of lymph node involvement			
None	35	27.56	-
1	61	48.03	-
2	31	24.41	-
Histological grade			
Grade 1	18	14.17	-
Grade 2	94	74.02	-
Grade 3	15	11.81	-
Metastasis			
Yes	36	28.35	-
No	91	71.65	-
Stage			
I, II, or IIIA	71	55.91	-
IIIB, IV, or IVB	56	44.09	-
TNM stage			
I/II	11	8.66	-
III/IV	116	91.34	-
Primary tumor resection			
Yes	119	93.70	-
No	8	6.30	-
Adjuvant chemotherapy			
Fluoropyrimidines	35	27.56	-
Fluoropyrimidines + Oxiplatin	70	55.12	-
Fluoropyrimidines + Biological drugs	19	14.96	-
None	3	2.36	-
Survival			
PFS	127	-	73.3 (41.0-NR)
OS	127	-	82.3 (62.1-NR)

ADC: adenocarcinoma; CRC: colorectal cancer; ECOG: Eastern Cooperative Oncology Group; OS: overall survival; PFS: progression-free survival; TNM: Classification of Malignant Tumors. Qualitative variables are shown as number (percentage, %). Quantitative variables with a normal distribution are shown as mean ± standard deviation. Quantitative variables with a non-normal distribution are shown as p_50_ (p_25_–p_75_).

**Table 3 cancers-15-04077-t003:** Influence of gene polymorphisms and clinical–pathological characteristics on overall survival of 127 CRC patients.

	Overall Survival	
Independent Variables	HR (CI_95%_)	*p*-Value
Family history of CRC (No)	1.442 (0.268–1.792)	0.449
Stage (IIIB, IV, or IVB)	1.130 (0.311–4.104)	0.852
Adjuvant chemotherapy (Fluoropyrimidines)	1.455 (0.595–3.558)	0.411
Adjuvant chemotherapy (Fluoropyrimidines + biologics drugs)	4.248 (1.655–10.904)	**0.003**
Adjuvant chemotherapy (None)	5.148 (0.507–52.332)	0.166
Lymph node involvement (Yes)	1.503 (0.751–3.006)	0.249
ECOG score (2–4)	1.342 (0.505–3.568)	0.555
Metastasis (yes)	1.360 (0.335–5.548)	0.664
Age of CRC diagnosis	1.067 (1.029–1.107)	**<0.001**
*CYP24A1* rs6068816 (TT vs. C)	6.237 (2.297–16.936)	**<0.001**
*GC* rs7041 (C vs. AA)	4.939 (1.578–15.457)	**0.006**
*VDR* rs7975232 (ApaI) (AA vs. C)	1.974 (1.045–3.731)	**0.036**

*p*-value (log-rank) < 0.001. HR: hazard ratio; CI_95%_: 95% confidence interval. Bold means the result is significant.

**Table 4 cancers-15-04077-t004:** Influence of gene polymorphisms and clinical–pathological characteristics on progression-free survival of 127 CRC patients.

	Overall Survival	
Independent Variables	HR (CI_95%_)	*p*-Value
Age of CRC diagnosis	1.055 (1.023–1.090)	**0.001**
ECOG score (2–4)	1.029 (0.412–2.577)	0.950
Lymph node involvement (Yes)	1.994 (0.982–4.049)	**0.056**
Metastasis (yes)	3.142 (1.096–9.004)	**0.033**
Stage (IIIB, IV, or IVB)	1.268 (0.452–3.556)	0.651
Primary tumor resection (No)	1.271 (0.311–1.992)	0.613
Adjuvant chemotherapy (Fluoropyrimidines)	2.306 (1.059–5.019)	**0.035**
Adjuvant chemotherapy (Fluoropyrimidines + biologics drugs)	1.861 (0.823–4.208)	0.135
Adjuvant chemotherapy (None)	1.870 (0.231–15.143)	0.557
*CYP24A1* rs6068816 (TT vs. C)	6.095 (2.413–15.398)	**<0.001**
*GC* rs7041 (A vs. CC)	1.543 (1.143–1.935)	0.268
*VDR* rs2228570 (FokI) (G vs. AA)	1.003 (0.412–2.420)	0.999
*VDR* rs7975232 (ApaI) (AA vs. C)	1.692 (1.402–2.092)	**0.066**

*p*-value (log-rank) < 0.001. HR: hazard ratio; CI_95%_: 95% confidence interval. Bold means the result is significant.

## Data Availability

Data unavailable due to privacy and ethical restrictions.

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
