# Peer review of "Single Nucleotide Polymorphisms in the Vitamin D Metabolic Pathway as Survival Biomarkers in Colorectal Cancer"

_cancers, 2023, doi:10.3390/cancers15164077_

Round 1

Reviewer 1 Report

In this article, Pérez-Duran et al. analyze the association of several polymorphisms in genes related to vitamin D metabolism with PFS and OS in a relatively small cohort of CRC patients. The results presented are of interest, but the study shows some major shortcomings that should be solved prior to publication 

Major comments 

1. Median follow-up is a key parameter in survival studies, but is not provided by the authors. 

2. According to methods, patients were recruited in 2019-2022 and followed up until February 2023. Consequently, the maximum follow-up for a patient would be 4 years (48 months). However, all over the results section (including figures), the authors present survival times systematically superior to 48 months and up to 140 months. This is inconsistent 

3. According to 3.2.1, several factors were significantly associated with OS in the univariate analysis; namely family history, age, ECOG, presence of metastasis, lymph node involvement and type of chemotherapy. However, only the presence of metastasis and age were included in the multivariate analysis presented in Table 3. This is incorrect. All factors found in the univariate should be included in the multivariate 

4. Same for PFS, all factors found in 3.2.2 should be included in the multivariate presented in Table 4 

5. Numbers at risk should be included below the Kaplan-Meier plots 

6. The scientific literature is filled with markers with supposed diagnostic or prognostic value that have never been validated. For a biomarker to be used in the clinic, validation in an independent cohort is an absolute necessity. The authors should stress this point in the discussion 

Minor comments 

1. The first sentence of the simple summary makes little sense and is poorly written. It should be removed 

2. In the introduction, a supplementary figure summarizing all the pathways explained in the third paragraph would facilitate reading 

3. The definition of PFS is incomplete. The authors should clarify if PFS was calculated from the time of surgery (surgical patients) or first line therapy (unresectable patients) to progression or death from any cause 

4. Some parameters are incorrectly listed in Table 2. The authors should provide median (not mean) age at diagnosis or, even better, the number of patients in 3-4 age categories (i.e., <45, 45-60, >60). Same for tumor size and histological grade. 

5. The authors should present the Kaplain-Meier plots of the entire cohort as a supplementary figure, not just the mean/median. And “standard deviation” is never used for survival since it does not consider censored patients due to end of follow-up 

6. Conclusions. The polymorphisms might associate with survival, but the authors cannot conclude that they are “responsible for lower OS and PFS”. An extensive set of biological experiments would be needed to demonstrate that.

English should be revised to avoid expressions such as “has gathered the attention”, “taking the points set out above”, “so much so”, etc

Reviewer 2 Report

In the manuscript, the authors report that they successfully conducted an observational retrospective cohort study. The authors evaluated the influence of 13 SNPs in the vitamin D metabolic pathway on colorectal cancer survival by evaluating progression-free survival and overall survival. The research indicates that the SNPs involved in the vitamin D metabolic pathway may have potential utility as prognostic biomarkers of colorectal cancer. But they still need to be improved as mentioned below.

1.      What are the criteria for selecting Gene SNPs in the detection of genetic polymorphisms?

2.      The influence of clinicopathological characteristics and gene polymorphisms on survival should be better integrated and analyzed.

3.      Impact on the survival between different SNPs interaction should be considered.

4.      The Kaplan-Meier plot of the different genotypes of each SNP can be presented in a single chart for comparison, as shown in Figure 4 and Figure 5.

5.      In the discussion section, the author should try to explain the reasons why the results of this study are not completely consistent with those of previous studies.

6.      The effect of SNPs genotypes with survival marker function on survival should be listed in the conclusion.

Minor editing of English language required.

Round 2

Reviewer 1 Report

All my comments have been satisfactorily answered

English needs some improvement